# Assessing concentration in the monoclonal antibody innovation market: A patent-based study

**André Soares Motta-Santos**[1,2]*, **Leonardo Costa Ribeiro**[1], **Jeff Gow**[2], **Khorshed Alam**[2], **Kenya Valéria Micaela de Souza Noronha**[1], **Mônica Viegas Andrade**[1]

**1** Center for Development and Regional Planning (CEDEPLAR), Universidade Federal de Minas Gerais (UFMG), Belo Horizonte, Minas Gerais, Brazil, **2** School of Business, University of Southern Queensland (UniSQ), Toowoomba, Queensland, Australia

* andresantos111@ufmg.br

## Abstract

### BACKGROUND

Monoclonal antibodies (mAbs) are revolutionizing healthcare treatments due to their high efficacy and relative safety, despite their cost. Since they first appeared in the late 1980s, a rapidly growing market has developed.

### OBJECTIVE

This study aims to analyze concentration levels in the market for mAb innovations through a quantitative patent analysis. Data were analyzed using traditional concentration indicators such as the Herfindahl-Hirschman Index and Concentration Ratio, as well as linear regression and kernel density graphs to evaluate innovation and global technology dissemination strategies. The starting point was patents associated with mAbs registered by the FDA and identified in the IQVIA database up until 2019, and supplemented by data from The Antibody Society, Purple Book, Orange Book, and FDA.

### RESULTS

Our findings indicate that the market for mAb innovations is moderately concentrated for general patents and unconcentrated for priority patents. However, it is significantly more concentrated than the market for chemical drug innovations. The mAb patent families tend to generate more progeny patents, although they are deposited in fewer countries. Chemical drug patents spread faster. Some companies seem to be central to the development of mAbs worldwide, including Roche, PDL, City of Hope, and Celltech. Other important players in the mAb innovation market are AbbVie, Amgen, Novartis, GSK, Biogen, BMS, Regeneron, J&J, and AstraZeneca. The most relevant patents in the analysis are associated with methods and procedures to obtain mAbs, not with molecules themselves.

provided the original author and source are credited.

**Data availability statement:** We created our own dataset base on other data sources, as detailed in the article. Some of the data is publicly available, but most restrictions apply to other kinds of data, as explained below. The Orange Book (https://www.fda. gov/drugs/drug-approvals-and-databases/ approved-drug-products-therapeutic-equiv- alence-evaluations-orange-book) and Purple Book (https://purplebooksearch. fda.gov/) databases are publicly available. Data taken from the FDA are available at the FDALabel website (https://www.fda.gov/ science-research/bioinformatics-tools/fdala- bel-full-text-search-drug-product-labeling). The list of mAbs is available at the Antibody Society website (https://www.antibodysociety. org/resources/approved-antibodies/). General patent information from USPTO is also publicly available (https://ppubs.uspto.gov/pubwebapp/ static/pages/landing.html). Patent informa- tion and related data pertaining to PATSTAT and IQVIA is paywalled and not in the public domain. This part of the data underlying the results presented in the study are available from these parties' websites, respectively at: PATSTAT: https://www.epo.org/en/search- ing-for-patents/business/patstat IQVIA: https:// www.iqvia.com/.

**Funding:** The author(s) received no specific funding for this work.;

**Competing interests:** The authors have declared that no competing interests exist.

## CONCLUSION

The concentration in the mAb innovation market is higher than the concentration in the market for chemical drugs innovations. Our findings also indicate that expertise in mAbs development and production is concentrated in a few countries. Additionally, our study identified that a few key players from high-income countries are driving innovation in the mAb market.

## Introduction

Healthcare is highly dependent on innovation. New technologies have improved the diagnosis, prevention, and treatment of many diseases over the years, but usually at a high economic cost [1–3]. The fast-growing pace of the pharmaceutical industry has intensi- fied pressure on stakeholders to deliver new technologies to meet people's needs as rapidly and safely as possible. One of the latest developments in the pharmaceutical industry is associated with targeted therapies, including monoclonal antibodies (mAbs). These are laboratory-created immune system proteins produced by a single B cell clone [4–6]. They act as substitute antibodies to restore, enhance or mimic the immune system response to specific antigens [7,8]. Their uses for hematologic malignancies, solid tumors, and autoim- mune disorders are the most widely known [6,9–11]. Other illnesses have been targeted by these technologies, including metabolic, infectious and ophthalmologic diseases, as well as drug reversal [4,12–14]. They have also been instrumental in the treatment of patients with severe COVID-19 [15,16]. Recently, the development of antibody-drug conjugates (ADCs) and bispecific antibodies has widened even more the usefulness of mAb technology, espe- cially for cancer treatments [17–22]. In the short term, continued investment in R&D for new mAb technologies could provide the world with technologies to treat or cure many rare and orphan diseases [5,6,8,23,24].

The onset of mAb research happened in 1975, when Milstein and Köhler [25] described a method for obtaining specific antibodies from a continuously growing cell line. In 1986, after some additional advances, the first monoclonal antibody (mAb), Muromonab-CD3, was approved for human use by the Food and Drugs Administration (FDA) [7,14]. How- ever, it was only after the development of the methods for obtaining chimeric, humanized, and human mAbs that the new molecules start getting in the market at an increasing pace [4,26,27]. mAbs are associated with substantial research and development (R&D) costs. According to DiMasi and Grabowski [28], the average capital cost to develop a new biophar- maceutical drug was approximately US$1.2 billion during the 2010s. Despite these significant expenses, mAbs high prices consistently translate into abundant revenues for pharmaceutical companies, making them a lucrative investment [29–31].

Many mAbs and the associated processes are relatively recent and patent-protected. According to the TRIPS agreement, the standard patent protection term for pharmaceutical technologies is 20 years, but this period can be extended when new, relevant uses are discov- ered [32,33]. To file a patent, innovators must disclose information on the technology, inven- tors, and holders. Therefore, patent registries are a good, reliable source of data for studying innovation [8,34–39]. Patent analytics has been used to describe and analyze (i) the research and innovation process in national and international contexts [34,40–43], (ii) the level of technology development in a particular sector [36,41,44], (iii) the interdependence between industrial sectors and technology fields [45], and (iv) the technological capacity at country, sector, or company levels [46]. Previous studies have applied patent analytics to examine the market for mAb innovations, primarily using descriptive approaches [15,47,48]. These studies

indicate that the mAb innovation market is dynamic and dominated by a few major players, mostly from high-income countries (HICs). The United States (US) is the largest contributor of market players and scientific publications on this subject worldwide [8,11].

In the pharmaceutical market, three related processes – R&D, production, and commercialization – could be carried out by different companies for the same product, leading to three different estimates of market concentration. Specifically, for the drug production market, there is some evidence on concentration levels [49,50]. It was observed that the market as a whole is not concentrated enough to be considered an oligopoly. Craig and Malek [49] found that the five largest pharmaceutical companies retained only 15% of the market in the 1990s [49]. Information on the innovation market, especially mAbs, is much scarcer. This study focuses on R&D by analyzing patent deposit activities, with the specific aim of measuring concentration in the mAbs innovation market using an unprecedent database. Understanding the interactions among players and concentration in the innovation market is essential for developing policies to address the current high prices of mAbs. In addition, the dynamism of the field requires up-to-date analyses. To the best of our knowledge, this study is the first comprehensive analysis of mAbs innovation market.

## Methods

This is a cross-sectional patent analysis. An unprecedented dataset was developed utilizing several databases including information on patents, drugs, and inventors of the market for mAbs and chemical drugs. This study did not use primary data or data from individuals. Before reporting the methods and indicators used to measure and characterize market concentration, it is crucial to define the innovation market.

### Market definition

Markets must be defined with respect to their geographic and product dimensions. The geographic dimension comprises the territorial boundaries of the market; i.e., all relevant sources of the product and their spatial disposition. Since the formation of large drug companies in the last century, the borders of pharmaceutical markets have expanded [49]. Despite some trade restrictions between countries imposed by the TRIPS Agreement, the pharmaceutical production market has a global scope in the sense that commercialization of drugs might happen across frontiers [33,51,52]. The same is true for the innovation market. The flow of information occurs across borders through the allocation and enforcement of intellectual property rights (IPRs) in offices in each country. However, patent legislation *per se* are contained inside the borders of national innovation systems. The dependency of international products subjected to IPR protection presents two challenges for accessing new drugs, especially in LMICs: (i) the monopoly created by the IPR hinders competition and lead to higher prices and (ii) the logistics of distributing a new technology worldwide is difficult. Even registering new technologies with local authorities might delay the availability of new drugs.

The product dimension aspect aims to include all products that can be considered relevant substitutes [53]. With respect to the product dimension, the definitions of production and distribution markets differ significantly from that of the innovation market for pharmaceutical products. The production market can be defined at different levels: (i) The first level describes the *pharmaceutical market* as a whole. Since drugs are not substitutes in most cases, this definition is usually unhelpful [54,55]. (ii) Some authors suggested that the *indication* could be a good way to define a market [54]. In this case, two products are considered competitors when used for the same purpose. This definition has the downside of joining different *drug classes* in the same market. A drug class might be defined by different criteria. One of the most common

criteria is the mechanism of action, which describes how a drug produces the intended effect on the body. Therefore, (iii) another possibility is to define the market by the *mechanism of action*. This definition would help avoid mixing drug classes but would still neglect that drugs could be substitutes in one indication but not in others; that is, drugs in the same class can have many indications, but not all agents must have the same indications. (iv) To address this limitation, the market could be defined by *class-indication*. In this case, drugs in the same class would be competitors only if they fit the same purpose. However, in the pharmaceutical market, many products require prescriptions. This restriction leaves acceptable substitutes (me-too drugs and even drugs of different classes but similar indication) out of the competitive process. (v) The simplest way to define a market is by *agent-indication*. The level of aggregation depends on the objective of the analysis.

Unlike production and distribution markets, the product of the innovation market is knowledge. Knowledge is commonly subject to IPRs which are the tradeable products of innovation. The most common form in which they are accounted for in the pharmaceutical market is as *patents* [51]. The concept of pharmaceutical innovation includes not only knowledge about new drugs but also new procedures, techniques, and methods to achieve intermediate steps in the process of developing a new drug. Each patent describes a previously unknown technology, resulting in low cross-elasticity of the innovation market.

In this paper the pharmaceutical innovation market was divided into two categories: (i) biological drug patents, and (ii) chemical drug patents. The group of biological drugs is very heterogeneous, but, since this work is focusing on mAbs, their patents were chosen to represent the market for biological drug innovations. Chemical drug patents were used as the comparator. It is reasonable to assume that companies that have the know-how to develop a chemical drug could develop a me-too drug or copy another chemical drug, but that is not necessarily the case with a biological drug. The same should be true for companies that can produce mAbs, despite their much higher complexity. If a company has the capacity to develop mAbs, it could also develop biosimilars or me-too mAbs. For this study, the selected level of aggregation is sufficient to provide reasonably accurate concentration estimates.

## Database construction

**Fig 1** synthesizes the steps followed to build the database. The first step is the identification of the drugs. The list of mAbs approved by the FDA and their International Non-Proprietary Names (INNs) were collected from The Antibody Society website [56–58]. The Antibody Society is an international non-profit trade association representing individuals and organizations involved in antibody development. The list of INNs of chemical drugs approved by the FDA was collected from the Orange Book [59]. The Orange Book (Approved Drug Products with Therapeutic Equivalence Evaluations) identifies drug products approved by the FDA under the Federal Food, Drug, and Cosmetic Act. The INNs were used to retrieve the US Patent and Trademark Office (USPTO) patent numbers from the IQVIA database, which includes data from 130 countries [60].

Patent numbers were used to retrieve the patent metadata from the USPTO subset on PATSTAT. The USPTO is a federal agency responsible for granting patents and registering trademarks in the United States. The United States is a vast market that can be considered a good proxy for a global patent office [11,34,38]. PATSTAT is a database comprising almost 70 million patents from over 100 offices worldwide. It is organized by the European Patent Office (EPO) [34]. Data was obtained from PATSTAT up to 2019, as this is the cut-off point for the IQVIA database accessible for this study. The metadata retrieved at this point was the application number, country code, filing date, publication date, International Patent Classification

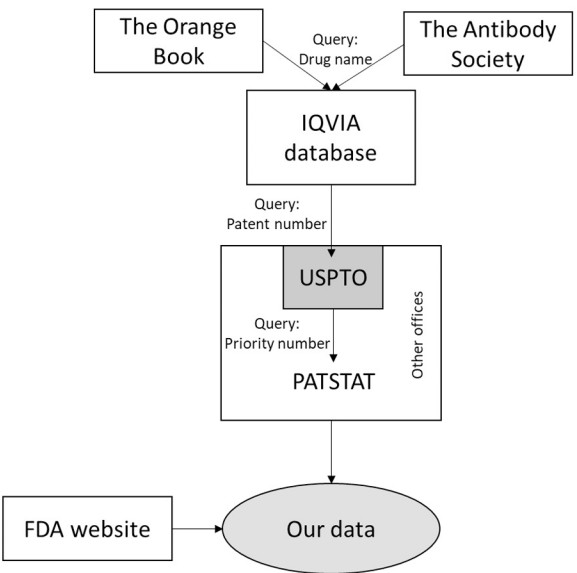

**Fig 1. Database construction steps.**

(IPC) codes, USPTO identification number, patent types, priority patent application numbers, and holders' names, countries, and types. The types of holders comprise (i) companies, (ii) individuals, (iii) government non-profit organizations (GNPOs), (iv) non-profit organizations (NPOs), (v) universities, and (vi) hospitals.

*Priority patents* are the first patent deposited to protect an invention worldwide [61]. The date of the priority deposit marks the beginning of IPR protection, after which inventors might seek protection in multiple countries. The set of multiple publications protecting the same technology in different countries is called a *patent family* [62]. The priority patent numbers were searched in PATSTAT (covering other patent offices), and their metadata and patent families were retrieved. Data obtained included IPC codes, application numbers, country codes, and filing dates. Finally, indications were extracted from the FDA website [63], Which provided information about approved drugs in the United States and their uses.

## Data analysis

Initially, we conducted a descriptive analysis of patent holders, including primary holders, types of holders, and patent offices. The holders were reclassified in the database to incorporate groups of companies and their subsidiaries (S1 Table). Three sets of indicators were used to examine the structure and strategies of the mAbs innovation market (S2 Table). First, traditional concentration indicators such as the *Herfindahl-Hirschman Index* (HHI) and the *Concentration Ratio* (CR) were used to determine the degree of market concentration in the mAb innovation market. Secondly, the strategies used by patent holders to wield their market power were investigated, specifically in terms of innovation and global technology dissemination. Lastly, the interdependence among patent holders in the mAb innovation process was analyzed. It was hypothesized that due to the intricate nature of the mAb innovation process, the intensity of these partnerships may potentially weaken the market power of individual holders. All analyzes were performed in R [64–66]. Details on the indicators are presented next, followed by the results.

## Market concentration

This paper applies the HHI and the CR ($CR_4$ and $CR_{10}$) to estimate market concentration. These indices are among the most well-established in the literature. Generally, they evaluate the distribution of market share, measured in our study with patents, among individuals or firms. Therefore, they estimate the competition in an industry [67]. The CR is the cumulative market share of the $M$ largest companies in the market (**Equation** 1) while the HHI is calculated by summing the squares of the market shares of all holders (**Equation** 2). Their values range from 0 to 1 (monopoly); more precisely, $D_{HHI|CR} = (0,1]$. A market can be considered unconcentrated when $HHI \leq 0.10$, moderately concentrated between 0.10 and 0.18, and highly concentrated when $HHI > 0.18$ [68–71]. For $I$ companies on a market, the HHI has its lowest value when all players have equal market shares; that is, $min(HHI) = {1}/{I}$ [67,70]. In the case of the CR, an industry is commonly considered competitive when $CR_4 < 0.4$ while $CR_4 > 0.6$ is associated with tight oligopolies or a market with a dominant firm with a competitive fringe [67].

$$CR_M = \sum_{m=1}^{M} \frac{N_m}{N^*} \tag{1}$$

$$HHI = \sum_{i=1}^{I} \left( \frac{N_i}{N^*} \right)^2 \tag{2}$$

A descriptive analysis of the participation of holders in general patents and priority patents was conducted. Subsequently, the HHI was calculated separately for general and priority patents in the mAb and chemical drug markets, and the results were compared. The HHI and CR values for patents indicate the market reserve around the world, while the values for priorities refer to knowledge generation, measuring concentration in technology development. A log-log graph of participation per holder was generated with estimated linear regression curves (**Equation 3**). Linear regression coefficients are associated with the degree of concentration. Higher absolute values of the coefficient (i.e., the slope of the log-log regression curve) indicate more concentrated markets. In addition, the coefficient of a log-log regression curve is equivalent to the elasticity, where the slope indicates that a 1% variation in the number of holders is associated with a $\beta_1$ variation in holder participation in total patents and priorities. Student's t tests were performed on the coefficients to evaluate statistical significance.

$$log(Y) = \beta_0 + \beta_1 log(X_1) + \varepsilon \tag{3}$$

## Strategies to wield market power

Two indicators were designed to reveal the strategies pharmaceutical companies use to wield market power. The first indicator identifies which types of holders are most innovative and which focus more on spreading, such as depositing the same patents in many countries as an expansion strategy. For each patent holder, a ratio (η) was calculated by dividing the number of priority patents by the number of general patents. Holders who focus on innovation have η close to one: the ratio between priority patents, which first protect an innovation, and general patents indicates that most of the company's portfolio is the first in their family. In contrast, players who focus on spreading have η close to zero. A kernel density graph was built to identify different strategies by holder types. The curves by holder type were then compared with each other for the mAb innovation market and between the mAb and chemical drug innovation markets.

The second indicator evaluates the pace of market protection. The fast spread of a patent family indicates the high formation of a market reserve. In addition, a family registered in only a few countries indicates that the technological capacity to replicate innovation is not widespread. Two measures were defined: the time from the deposit of the priority to the deposit of the patents in other offices by holder type, and the number of different countries in which progeny patents were registered, classified by holder type. The analyses were presented in cumulative distribution function graphs.

## Interdependence among holders

A descriptive analysis was used to demonstrate holder partnerships in R&D. The development of complex molecules such as mAbs usually requires transdisciplinary knowledge, leading to expected partnerships between companies and other institutions [8,11]. The inclusion of multiple holders in a patent leads to more bureaucratic procedures in the approval of 'progeny' patents, trade activities, and royalty distribution. The indicator was calculated by aggregating the number of holders per patent and priority. The data were sorted in decreasing order by the number of holders. Then a log-log graph and a linear regression of the log-log graph of the number of holders per patent were presented. The linear regression allowed for the comparison of the coefficients for each curve. In this case, the more distributed the number of holders per patent, the lower the coefficient. Student's t tests were performed on coefficients to test for statistical differences, with a significance level of 5%. In addition, cross-holder partnerships were described to understand whether more than one type of shareholder is commonly involved in the development of pharmaceutical technologies. A $\chi^2$ test was carried out on two matrices: one comparing the number of general patents of mAbs and chemical drugs by the number of different holder types and the other comparing priority patents.

Finally, a network linking molecules and priority patent holders was built. This network shows the most relevant holders, since a molecule might be associated with multiple priorities. Individuals were not included in the analysis because they are not vital to understanding the main players in the market and would make the visualization of the network more difficult. The metric applied to evaluate the power of a node was the *degree centrality* ($C_D$). Degree centrality refers to the number of direct edges that a node has (**Equation** 4, in which $G_{ij}$ is the value of the link between nodes $i$ and $j$ (0 or 1), and $n$ is the number of nodes in the network). Holders with a high degree centrality are responsible for essential patents for developing new drugs.

$$C_D(i) = \sum_{i=1}^{n} G_{ij} \qquad (4)$$

## Results

A total of 1,732 mAbs general patents and 4,636 chemical general patents were included in the analysis. There were 928 different holders for 1,703 general patents for mAbs in the database. The distribution of holder types shows that individuals were the most common holders (91.4%), followed by companies (5.5%) and universities (1.9%). Given that a holder can be associated with various patents and the same patent can have many holders, the distribution of patents per holder type shows that companies were associated with the largest number of patents (1,346, 79.0%), followed by individuals (644, 37.8%) and universities (115, 6.6%). mAb priority patents follow a similar pattern. Data on the holders of 445 out of 718 mAb priorities were available. Of the 241 different holders in the database, most were individuals (185, 76.8%), followed by companies (32, 13.3%) and universities (14, 5.8%). In this case, companies

were also associated with the highest number of patents (274, 61.6%), followed by individuals (120, 27.0%) and universities (71, 16.0%).

For chemical drugs, 4,580 holders were identified for 4,608 patents. Most of them were individuals (4,064, 88.7%), followed by companies (429, 9.4%) and universities (53, 1.2%). As with mAbs, companies held the highest number of patents (3,905, 84.7%), followed by individuals (2,117, 45.9%) and universities (179, 3.9%). Data on holders of 827 chemical priority patents were available. Of the 741 holders, most were individuals (478, 64.5%), companies (241, 32.5%) and universities (14, 1.9%). The distribution of patents among holders shows that companies are the most prevalent holder type (706, 85.4%), followed by individuals (252, 30.5%) and universities (24, 2.9%). It is evident that the market is dominated by companies, although some individuals might play a role in developing new drugs. Various partnerships were possible for the development and spread of new technology (S3 Table).

## Market concentration

The HHIs values for the mAbs innovation market were 0.105 for general patents and 0.077 for priority patents, characterizing moderately concentrated and unconcentrated markets, respectively. The HHIs for the chemical drug innovation market were much lower (0.010 and 0.015 for patents and priorities, respectively), indicating unconcentrated markets [71]. The $CR_{10}$ and $CR_4$ were, respectively, 0.677 and 0.498 for mAb patents, 0.683 and 0.438 for mAb priorities, 0.213 and 0.117 for chemical drug patents, and 0.254 and 0.133 for chemical drug priorities. Despite the relatively low HHI, the CRs show that the main players control a significant portion of the market of mAbs. The analysis of HHI and CR excluding individuals shows a similar pattern but with higher estimates, especially for mAbs (**Table 1**, S4 Fig).

The log-log curves of the participation profile are shown in **Fig 2**. The coefficients of the log-log regressions are larger in absolute terms for the mAb innovation market (0.94 and 0.90 for general patents and priorities) than for the market for chemical drug innovations (0.72 and 0.65). Hence, the market for mAbs presents a significantly more inclined curve (more concentrated) for general patents ($p < 0.001$) and priority patents ($p < 0.001$) than the market for chemical drug innovations, confirming the findings of the HHI and CR estimators. The general patent curves are also more inclined than the priority patent curves for mAb ($-0.94$ vs. $-0.90$, $p = 0.004$) and chemical drug innovations ($-0.72$ vs. $-0.65$, $p < 0.001$). As explained above, these coefficients represent the elasticities of each estimate. Therefore, the negative signs mean that the increase in the number of holders is associated with a lower participation of holders in general patents and priorities, which is a logical conclusion; i.e., the larger the number of holders, the more competitive (or less concentrated) the market is. A 1% increase in the number of holders is associated with a 0.94% decrease in participation for general mAb patents, a 0.90% decrease for mAb priority patents, a 0.72% decrease for general chemical

**Table 1. Concentration level estimates for the mAbs and chemical drug innovations markets.**

| Parameter | Market | All holders | | No individuals | |
|---|---|---|---|---|---|
| | | General Patents | Priority patents | General Patents | Priority patents |
| HHI | mAbs | 0.105 | 0.077 | 0.126 | 0.095 |
| | Chem | 0.010 | 0.015 | 0.011 | 0.015 |
| $CR_4$ | mAbs | 0.498 | 0.438 | 0.564 | 0.515 |
| | Chem | 0.117 | 0.133 | 0.133 | 0.150 |
| $CR_{10}$ | mAbs | 0.677 | 0.683 | 0.767 | 0.772 |
| | Chem | 0.214 | 0.254 | 0.242 | 0.284 |

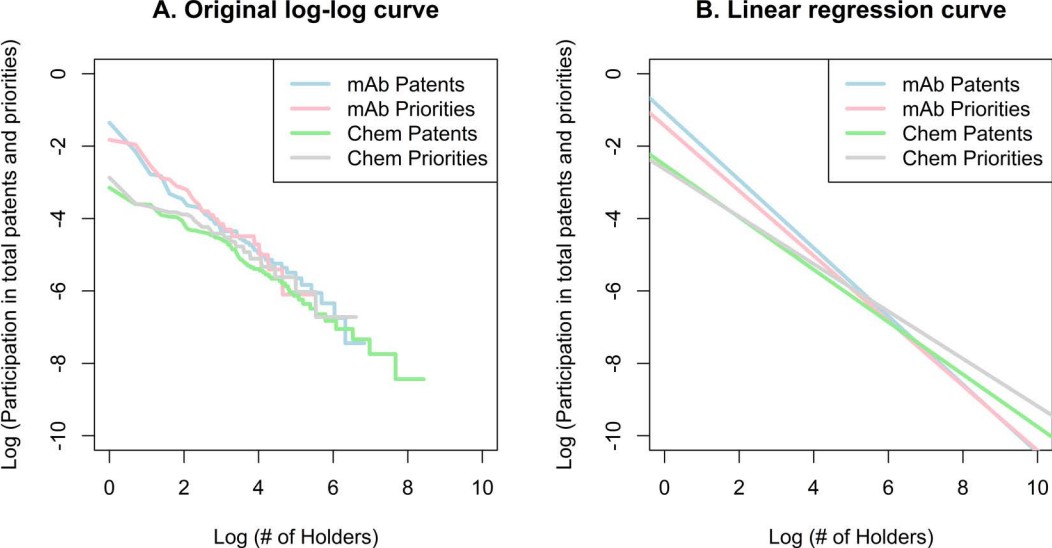

**Fig 2. Log-log estimation of the holder's participation in total patents and priorities.**

drug general patents, and a 0.65% decrease for chemical drug priorities. In summary, the market for mAb innovations is more concentrated and shows greater responsiveness to changes in the number of patent holders compared to the market for chemical drug innovations.

## Strategies to wield market power

**Fig 3** shows the different ratios between the number of priorities and general patents per holder type in the mAb and chemical drug innovation markets. In the mAb innovation market (**Fig 3A**), most companies and individuals focus more on spreading than developing, as indicated by their curves starting high (close to η=0), and decreasing. Both also show a small increase near the end (η=1), suggesting that some companies focus on innovation. GNPOs, hospitals, and universities have very different curves, starting lower in the density function and ending higher, indicating that these holder types are relatively more interested in developing new technology than spreading it. The market for chemical drug innovations is very different (**Fig 3B**). All curves seem to start in a higher position (η=0) and end in a lower position (η=1), suggesting that most players focus on spreading technologies while a smaller fraction focuses on developing for all holder types.

mAb priorities generated an average of 33.2 (s = 24.3) progeny patents worldwide. The average time between the first and last patent deposit in the same family was 10.4 (s = 5.8) years. The chemical drug priorities generated an average of 30.1 (s = 26.1) progeny patents, with an average time between the first and last deposits of 7.7 (s = 4.6) years. mAb priorities generated more progeny (p = 0.027) that were deposited over a longer period (p < 0.001). **Fig 4** compares mAbs and chemical drugs with respect to the time from the deposit of priorities and general patents. Patent filing is faster in the market for chemical drugs for companies, individuals, hospitals, and in general. For universities and GNPOs, the curves are intertwined but still achieve 100% faster for chemical drugs than mAbs. Although the chemical drugs innovation market is associated with fewer progeny patents on average, it has a larger interval than the mAb market (1 to 41 vs. 1 to 54). It is also associated with a higher average number of countries [10.6 (s = 9.1) vs. 12.3 (s = 11.7), p = 0.47], but not significantly. The number of progeny patents and the number of countries in which progeny patents are deposited are not

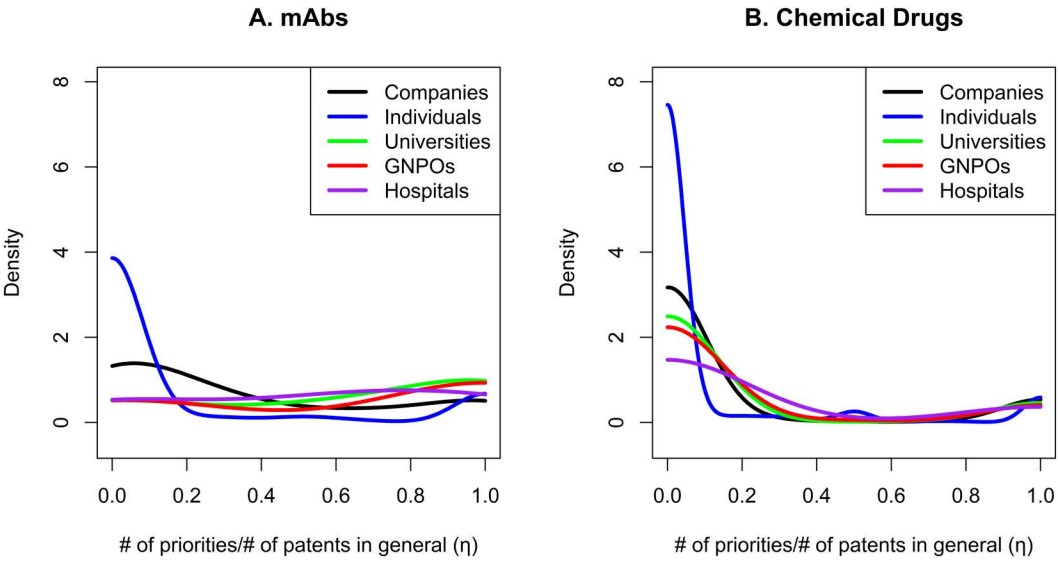

**Fig 3. Participation of different holder types in technology development.** Fig 3A = participation of different holder types in technology development in the market for mAbs; Fig 3B = participation of different holder types in technology development in the market for chemical drug.

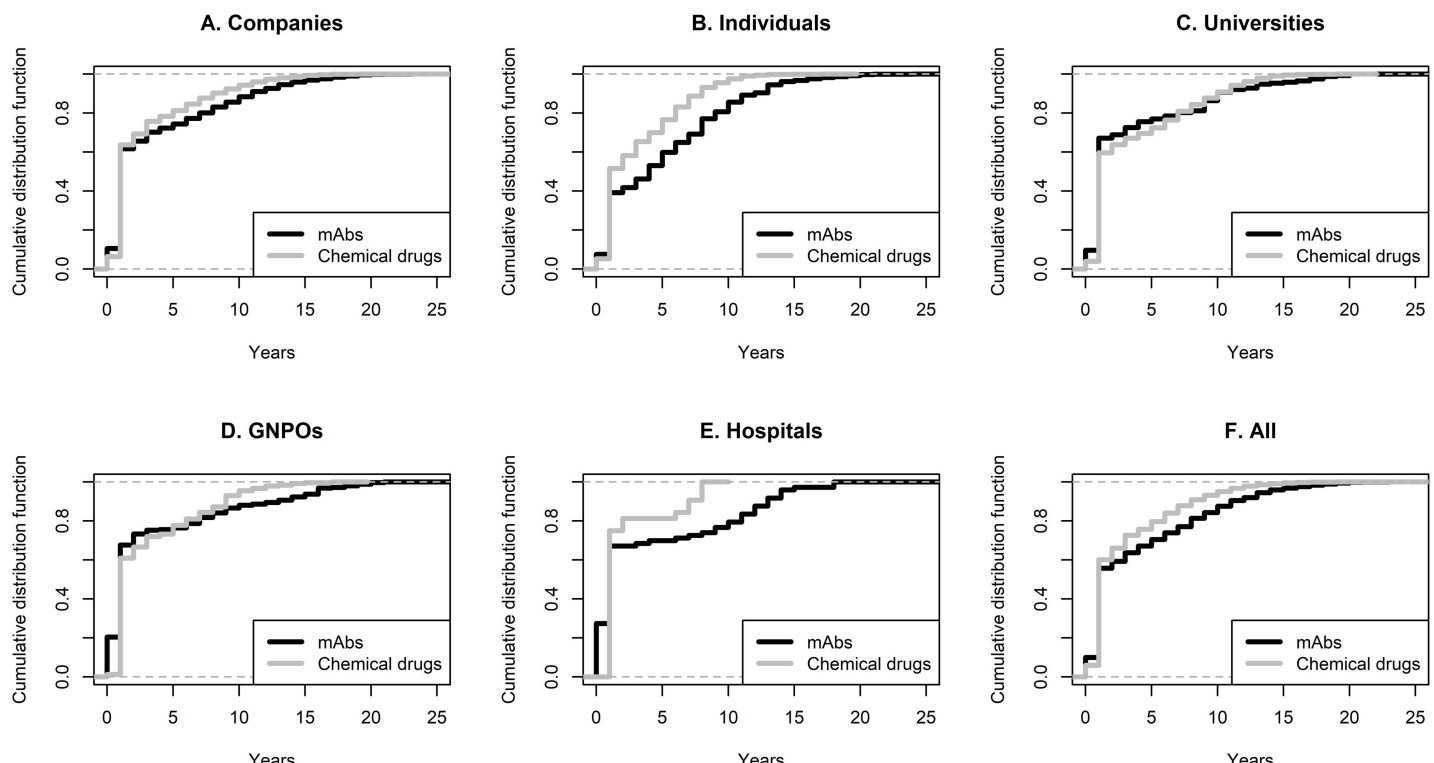

**Fig 4. Cumulative distribution functions comparing the market for mAb and chemical drug innovations regarding the time between the priority and progeny patents deposit.**

the same. A priority can have more than one progeny patent in the same office, which is more common in the mAb market, explaining the difference in the two analyses.

## Interdependence among holders

The average number of general and priority patent holders in the mAb innovation market was estimated at 2.68 (s = 3.06) and 1.84 (s = 1.93), respectively. The same analysis for the chemical drug market returned values of 2.69 (s = 2.80) and 1.88 (s = 1.71). The differences are not statistically significant between markets for general patents (p = 0.89) or priority patents (p = 0.70). However, mAb and chemical drug priorities have, on average, fewer holders than general patents ($\bar{x}$ = 2.68 vs. $\bar{x}$ = 1.84, p < 0.001 and $\bar{x}$ = 2.69 vs. $\bar{x}$ = 1.88, p < 0.001, respectively). This is an indication that fewer holders are associated with the innovative process than with market protection for each innovation.

The log-log graph and the regression line provide additional information ([Fig 5]). General patents have more inclined curves than priority patents for mAb (−0.76 vs. −0.58, p < 0.001) and chemical drug innovations (−0.74 vs. −0.59, p < 0.001). This observation suggests that the number of holders for priority patents is better distributed than for general patents. General patent curves are also above priority curves, which suggests, again, that priorities are usually associated with a lower number of holders. The difference in the coefficients of the linear regressions is very small for mAb vs. chemical drug patents (−0.76 vs. −0.74, p = 0.002) and mAb vs. chemical drug priorities (−0.58 vs. −0.59, p = 0.32), suggesting that the curves are almost parallel. Despite the significance in the patent coefficients, no substantial difference in terms of concentration should be concluded.

The cross-holder-type partnerships associated with developing mAb patents are lower than those in the chemical drug market. Approximately 72% of the mAbs and 65% of the chemical drug general patents were deposited by a single type of holder. Two or more types of holders were responsible for 28% and 35% of such patents, respectively. A similar pattern was observed for the priorities, which is even more surprising. In this case, 84% of mAb priorities

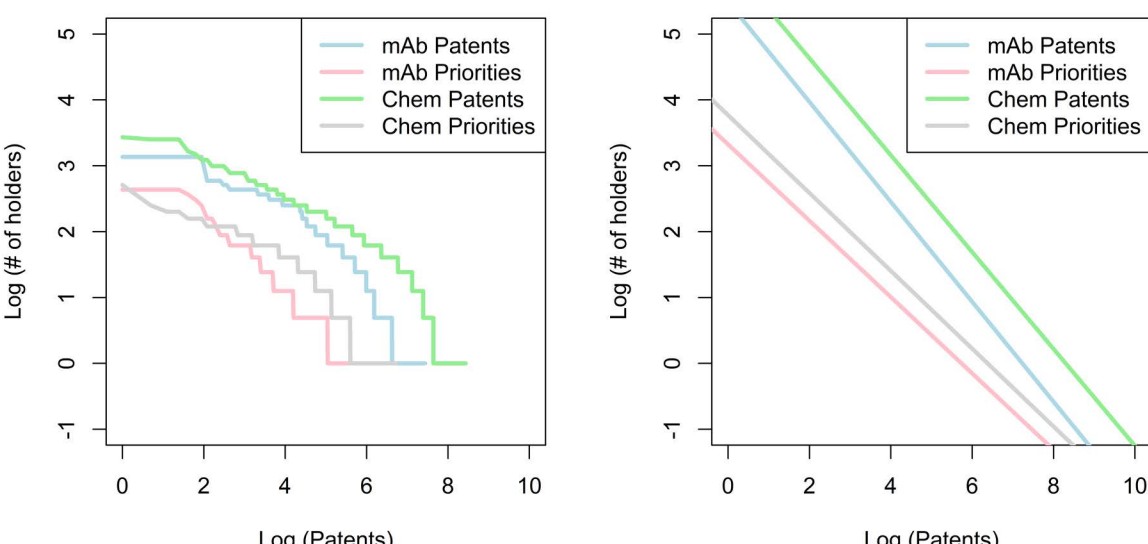

**Fig 5. Log-log estimation of the partnerships for the creation of patents and priorities.**

and 80% of chemical priorities were developed by a single-holder type, while 16% and 20% were associated with two or more holder types (Table 2). The estimate $\chi^2$ for the comparison of general patents and priorities between markets resulted in $p < 0.001$ and $p = 0.06$, respectively. These results suggest that the distribution of cross-holder-type partnerships differs between mAbs and chemical drugs. Despite their commonality, these partnerships are not the primary way for developing new drugs. The initial hypothesis that the cross-holder-type partnerships were paramount to the development of mAbs may not be completely correct. Other factors might play a role in the development of new drugs, such as companies developing knowledge independently after an initial catch-up period or acquiring IPRs from other innovators. Partnerships might make licensing and agreements between companies more complex and reduce gain from royalties, which would incentivize players to avoid them when possible.

A directed network was created to describe the holders responsible for the priorities cited by the mAbs. In Fig 6, red nodes represent institutions, and the green nodes represent molecules. The size of the sphere is associated with the degree of centrality. Roche (degree centrality = 29), City of Hope (23), PDL (15), and Celltech (14) are the most important institutions in terms of technology within mAbs. Adalimumab was the molecule most related to the priorities of different companies (12), followed by rituximab (10).

Each priority was included in the description of an average of 1.636 mAb (s = 2.06, min = 1, max = 24). The number of indications for the drugs produced by each priority was also

**Table 2. Number of holder types registered as patent and priority holders.**

| # of holder types | # of mAb patents | % of mAb patents in relation to total | # of chemical drug patents | % of chemical drugs patents in relation to total | # of mAb priorities | % of mAb priorities in relation to total | # of chemical drug priorities | % of chemical drugs priorities in relation to total |
|---|---|---|---|---|---|---|---|---|
| 1 | 1,219 | 72 | 2,985 | 65 | 375 | 84 | 662 | 80 |
| 2 | 471 | 28 | 1,606 | 35 | 69 | 16 | 165 | 20 |
| 3 | 13 | 1 | 16 | 0 | 1 | 0 | 0 | 0 |
| 4 | 0 | 0 | 1 | 0 | 0 | 0 | 0 | 0 |
| Total | 1,703 | 1 | 4,608 | 1 | 445 | 1 | 827 | 1 |

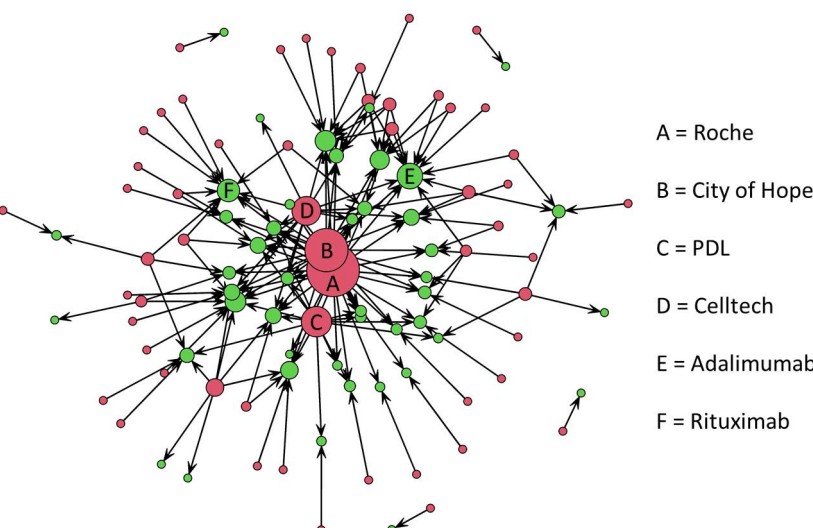

A = Roche

B = City of Hope

C = PDL

D = Celltech

E = Adalimumab

F = Rituximab

**Fig 6. Priorities associated with developing medicines according to the holder.**

evaluated. On average, the priorities were associated with 8.28 indications (s = 10.72). The ten most referred drug priorities were all held by companies (S5 Table). Many of these priorities protect methods rather than molecules or new uses, which makes sense. They were published between 1983 and 2003. The two priority patents associated with the highest number of drugs were published in 1988 (US6331415) and 1995 (US7923221) and were associated with 24 and 23 mAbs, respectively. The first describes "methods of producing immunoglobulins, vectors, and transformed host cells for use therein" [72], and the second describes "methods of making antibody heavy and light chains having specificity for a desired antigen" [73]. Genentech, a subsidiary of Roche, holds both. They are known as *Cabilly II* and *Cabilly III*. They expired in 2018. The third, fourth and fifth were held by PDL. The sixth was held by Roche and is known as *Cabilly I* [74]. PDL also held the last four.

Roche is a large multinational pharmaceutical company, still active and growing in the mAbs and chemical drugs markets. Protein Design Labs (PDL) was an American publicly traded holding that managed patents and other intellectual property generated by the company. PDL held seven of the top ten priorities in the list. They describe general methods for humanizing antibodies and obtaining humanized immunoglobulins. The company was a pioneer in the development of mAbs. In July 2020, PDL issued a statement of a Plan of Dissolution for the company [75].

## Discussion

The main findings of this paper show that the innovation market for mAbs is moderately concentrated for general patents and unconcentrated for priority patents. These concentrations are significantly higher than those observed in the chemical drug market for both general and priority patents (HHI = 0.105 and 0.077 vs. 0.010 and 0.015, respectively). The market share of the leading players further supports these conclusions. For instance, the $CR_4$ analysis reveals that the top four players hold 49.8% and 43.8% of general and priority mAbs patents, respectively, whereas these figures are comparatively lower for chemical drugs: 11.7% and 13.3%. This concentration of knowledge can affect industry performance and profitability, potentially leading to a loss of consumer surplus [70]. Higher concentration is associated with less competitive markets and, consequently, higher prices.

It is paramount to understand how the HHI and CR relate to each other. The relationship between them is not monotonic and depends on the market being evaluated [67]. They do not necessarily indicate the same result, but the analysis of both can lead to some conclusions. One inference that can be drawn from our concentration indices is that the mAb innovation market comprises a core of large companies surrounded by numerous smaller firms; i.e., an oligopoly with a competitive fringe. The high concentration in the mAb innovation market is a significant factor contributing to elevated drug prices. Estimates of HHI and $CR_4$ for different industries are numerous. Each industry has its own specificities and trends. For comparison, some of examples are: the US consumer book industry (HHI = 0.118, $CR_4$ = 0.368) [76], the global personal computers industry (HHI = 0.148, $CR_4$ = 0.707) [77], the Turkish automobile industry (HHI = 0.079, $CR_4$ = 0.462), and the biodiesel industry in Brazil (HHI = 0.059, $CR_4$ = 0.409) [78].

Despite the complications of copying mAbs, the market for biosimilars have been gaining traction over the years. Biological drugs are much more difficult to imitate than chemical drugs. From an economic perspective, the production of biosimilars involves more barriers to entry than the production of small molecules [79]. Specifically, in the case of mAbs, there is an essential differentiation of products. *Biosimilars* are not generic mAbs. They can be different with respect to the protein sequence and, thus, effects. The expiration of many mAb patents in the next few years should ignite the development of biosimilars. Remicade® (infliximab) sales

were slowed by 15% due to biosimilars between 2015 and 2017. A drop that is still modest, considering the prices of reference mAbs. Developing biosimilars that act better than reference products, also known as *biobetters* or *biosuperiors*, should accelerate this process [12] and help reduce the concentration in the market for production of mAbs. However, their imitative nature suggests they will not affect innovation market concentration.

The leading players in the mAb innovation market align closely with the dominant companies in the production market. Notably, several companies holding the highest number of patents (Roche, AbbVie, Amgen, Novartis, GSK, Biogen, BMS, Regeneron, J&J, and AstraZeneca) have also been recognized as major players in the production market. In 2017, seven companies dominated the mAb market, accounting for 87% of the sales. They were (i) Genentech (Roche Group) with 31% and 11 molecules; (ii) AbbVie with 20.0%; (iii) J&J with 13.6%; (iv) BMS with 6.5%; (v) Merck Sharp & Dohme (MSD) with 5.6%; (vi) Novartis with 5.5%; and (vii) Amgen with 4.9% [12]. One possible explanation for this high correlation among players in both the innovation and production mAb market is patent protection and the advantage of being a first-mover.

Different types of players employ distinct strategies in the mAbs market. Companies and individuals tend to focus on disseminating knowledge, while GNPOs and universities tend to prioritize the development of innovations. This difference in approach may stem from the limited production capacity of GNPOs and universities, which often choose to collaborate with pharmaceutical companies to facilitate technology application. In addition, the academic community was historically skeptical regarding patent protection, with many scientists believing that patents could slow scientific progress [6]. However, commercial interests have shifted this perspective over time. In contrast to the mAb market, the strategies of different stakeholders in the chemical drug market are less clearly defined, likely due to the market's maturity, lower complexity, and more competitive structure, with a larger number of players.

The number of holders is a proxy for the need of partnerships in the innovation process (priority patents) and market protection (general patents). There was no difference in the number of holders per general or priority patent when comparing the markets for mAbs innovations and chemical drug innovations. This result was unexpected, given the greater complexity of the innovation process for mAbs compared to chemical drugs. On average, priorities are held by fewer stakeholders than general patents for both the mAbs and chemical drugs innovation market. Therefore, fewer holders might be associated with the innovative process than with market protection for each innovation. Twenty-eight companies were associated with priorities but had no progeny patents in the USA. These companies may have developed the technology and negotiated their rights with other firms or participated in partnerships to spread the technology. Outsourcing might also play an important role in the development of new drugs.

Montalban and Sakinç [3] argued that the innovation process happens through networks of companies continuously motivated to expand the number of alliances. They showed that between 1980 and 2002, the number of alliances between large pharmaceutical companies and small biotechnology firms increased almost 20 times. Malerba and Orsenigo [55] suggested that new scientific methods and organizational strategies in the pharmaceutical industry have evolved over time, expanding partnerships with other organizations. Initially, some large corporations responded through mergers and acquisitions. Then, they started to increase their partnerships with small biotech companies, university spin-offs, startups, and academia. Eventually, companies began joint ventures and other agreements. The justification for this trend is that pharmaceutical companies do not have the knowledge and expertise necessary to overcome all obstacles to innovation [3,55]. However, our results suggest that this path is not the most common. It seems that partnerships between academia, spin-offs, startups, and companies could have increased everywhere, regardless of the type of drug being produced. The mAbs market is associated with fewer partnerships than the chemical drug market. The initial hypothesis that

the number of partnerships should be higher for mAbs because of their complexity seems false. One plausible explanation for these results is that larger companies acquire IPRs from smaller ones and dominate the market because of their investment portfolio and built-in capacity. It is possible that large companies form partnerships in some stages of the development of new drugs but eventually absorb the know-how and either drop or acquire the partner.

The findings related to the pace of market protection revealed different strategies between mAbs and chemical drug innovators. On average, mAbs priorities lead to the generation of more progeny than chemical drug priorities, which spread over longer periods. Knowledge in the chemical drug market seems to spread faster. Despite the lower number of progeny patents, the number of countries in which they are deposited is higher for chemical drug patents. One reason for this result is that evergreening might be more common in the mAb market, due to their high prices. *Evergreening* is the process of expanding a patent's lifetime by obtaining multiple patents covering slightly different aspects of the same invention [80]. Another reason is that many countries have the technological capability to produce generics of chemical drugs, but not many can copy a mAb. Therefore, developers of chemical drug technology might be more concerned with the time it takes for their inventions to be patented and in which countries. The size of the family and the delay from the first to the last publication might reflect the capacity of other producers to use the technology.

Network analysis shows that Roche Group, City of Hope, PDL, and Celltech are the most relevant institutions in the development of new mAbs and the intermediary processes to produce them (high-degree centrality in the directed network). The degree centrality quantifies the power of the node [81], i.e., how well connected it is. Therefore, these companies hold patents that are generally more important for the mAb market. The number of molecules associated with a priority indicates its usefulness and application. Companies with priority patents associated with more drugs tend to have more influence over the market. One example is Roche's *Cabilly patents*. Because Cabilly II is associated with a crucial stage of the mAb production process, significant manufacturers such as Abbott, J&J, ImClone, and MedImmune had to acquire licenses to use the knowledge. These patents have been the subject of many lawsuits over the years [82].

There are some concerns regarding high prices for medicines and the delayed entry of competitors in the market even more than 30 years after the Trade-Related Aspects of Intellectual Property Rights (TRIPS) Agreement was signed [32,33,40,55,79,83]. The high prices of medicines are derived from a complicated entanglement between entry barriers (such as high entry costs, high technology necessity, licensing, promotion, and patent protection), concentration, and market structures. Some authors state that the high prices of medicines lead to catastrophic family expenses and lack of access [84,85]. However, pharmaceutical companies argue that substantial R&D investments are responsible for the high price of medicines, and reductions might compromise the development of new drugs [79,83]. The use of artificial intelligence and machine learning in the development of new drugs might eventually influence this landscape, leading to higher R&D productivity [86,87].

Companies may also use strategies to retain their market power, avoiding competition and increasing concentration levels. One of these strategies is *evergreening*. This practice has been associated with delays in the entry of new competitors in the market and increasing healthcare costs [80,88,89]. The Cabilly patents can be considered a continuation of each other, with their practical lifetime (from the priority filling date to the expiration date) extending to 35 years due to overlap [82]. A more complex problem is the extension of patents through *reverse payments* [79,90]. Pharmaceutical companies may want to retain market dominance by paying competitors not to enter the market until a specific date. The patent period should be sufficient to allow innovators to recover costs [9,27,91].

This work has several limitations. (i) The reclassification of holders in the database is complicated by many acquisitions and fusions over the years. We opted to describe the market as it currently stands. (ii) The use of HHI or CR for concentration is not informative of the relevance of each invention or market shares. A holder with a small number of patents can be more important than a holder with many less relevant patents. The most important patents in the pharmaceutical market are not related to products, but intermediary processes in the way to obtain them. The average number of patents associated with each mAb was ranged from 1 to 387 from diverse holders. Therefore, the relationship between patent concentration and market share in the production market is not direct. There are even companies that hold important patents and do not produce a single molecule. To mitigate this limitation, we conducted network analysis and examined the most frequently cited priorities. (iii) Our database includes data up to 2019, therefore, many new mAb patents and priorities have not been considered. Nevertheless, this time restriction does not diminish the value of creating a new database specifically for evaluating the pharmaceutical innovation market. (iv) The cross-elasticity in the pharmaceutical market is low; i.e., products and patents are not generally substitutes for each other. We opted to divide the market into mAbs and chemical drugs for proximity, but these analyses could have been performed considering different drug classes, fields, indications, or even molecules to find specific concentration estimates. However, the analyses of such subgroups are out of the scope of our work. It is first important to understand the market as a whole before we can discuss specific (sub)markets.

The main contribution of this paper is the analysis of the concentration within the pharmaceutical innovation market, using a dedicated database to examine the strategies companies employ to maintain market power. Our findings also indicate that expertise in mAbs development and production is concentrated in a few countries, leaving many regions without the necessary capabilities to innovate or manufacture mAbs. Additionally, our study identified key players driving innovation in the mAb market: Roche, PDL, City of Hope, and Celltech. Other significant contributors include AbbVie, Amgen, Novartis, GSK, Biogen, BMS, Regeneron, J&J, and AstraZeneca.

A graphical abstract of this work is available at S6 Fig.

## Supporting information

**S1 Table. Grouping of companies for analyzing monoclonal antibodies patent holders.** (DOCX)

**S2 Table. List of indicators used in the analysis and their economic relevance.** (DOCX)

**S3 Table. Patent and priority holders by type and partnerships.** GNPO = Government Non-Profit Organization; NGO = Non-Government Organization. (DOCX)

**S4 Fig. Concentration level estimates for the mAbs and chemical drug innovations markets.** HHI = Herfindahl-Hirschman Index; CR = Concentration Ratio (TIF)

**S5 Table.** mAb associated priorities and their indications # = Number; US = United States of America (DOCX)

**S6 Fig. Graphical abstract.** (TIF)

## Acknowledgements

Dr. Motta-Santos acknowledges the *Coordenação de Aperfeiçoamento de Pessoal de Nível Superior* (CAPES) and University of Southern Queensland (UniSQ) for his scholarships. Dr. Andrade acknowledges the National Council for Scientific and Technological Development (CNPq) for her research productivity scholarships (processes 305592/2017-3 and 309252/2021-0). Dr. Noronha acknowledges CNPq and Minas Gerais State Agency for Research and Development (FAPEMIG) for her research scholarships (304632/2019-8 and PPM-00604-17, respectively). Dr. Ribeiro acknowledges CNPq and CAPES for his research scholarships (312020/2021-0 and 88887.837596/2023-00, respectively).

## Author contributions

**Conceptualization:** André Soares Motta-Santos, Leonardo Costa Ribeiro, Jeff Gow, Khorshed Alam, Kenya Valéria Micaela de Souza Noronha, Mônica Viegas Andrade.

**Data curation:** André Soares Motta-Santos.

**Formal analysis:** André Soares Motta-Santos, Leonardo Costa Ribeiro, Jeff Gow, Khorshed Alam, Kenya Valéria Micaela de Souza Noronha, Mônica Viegas Andrade.

**Investigation:** André Soares Motta-Santos.

**Methodology:** André Soares Motta-Santos.

**Project administration:** Leonardo Costa Ribeiro, Jeff Gow, Khorshed Alam, Kenya Valéria Micaela de Souza Noronha, Mônica Viegas Andrade.

**Supervision:** Leonardo Costa Ribeiro, Jeff Gow, Khorshed Alam, Kenya Valéria Micaela de Souza Noronha, Mônica Viegas Andrade.

**Writing – original draft:** André Soares Motta-Santos, Leonardo Costa Ribeiro, Kenya Valéria Micaela de Souza Noronha, Mônica Viegas Andrade.

**Writing – review & editing:** André Soares Motta-Santos, Leonardo Costa Ribeiro, Jeff Gow, Khorshed Alam, Kenya Valéria Micaela de Souza Noronha, Mônica Viegas Andrade.

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
