## [Decision Letter · Decision Letter 0]

5 Jan 2025

PONE-D-24-54902Assessing Concentration in the Monoclonal Antibody Innovation Market: A Patent-Based StudyPLOS ONE

Dear Dr.  Motta-Santos,

Thank you for submitting your manuscript to PLOS ONE. After careful consideration, we feel that it has merit but does not fully meet PLOS ONE’s publication criteria as it currently stands. Therefore, we invite you to submit a revised version of the manuscript that addresses the points raised during the review process.

We look forward to receiving your revised manuscript.

Kind regards,

Satish Rojekar, Ph.D.

Academic Editor

PLOS ONE

Journal Requirements:

Reviewers' comments:

Reviewer's Responses to Questions

**Comments to the Author**

1. Is the manuscript technically sound, and do the data support the conclusions?

Reviewer #1: Yes

Reviewer #2: Yes

Reviewer #3: Yes

Reviewer #4: Yes

2. Has the statistical analysis been performed appropriately and rigorously?

Reviewer #1: Yes

Reviewer #2: Yes

Reviewer #3: Yes

Reviewer #4: Yes

3. Have the authors made all data underlying the findings in their manuscript fully available?

Reviewer #1: Yes

Reviewer #2: Yes

Reviewer #3: Yes

Reviewer #4: Yes

4. Is the manuscript presented in an intelligible fashion and written in standard English?

Reviewer #1: Yes

Reviewer #2: Yes

Reviewer #3: Yes

Reviewer #4: Yes

5. Review Comments to the Author

Reviewer #1: Dear Author,

Assessing Concentration in the Monoclonal Antibody Innovation Market: A Patent Based Study is interesting article on the large molecules which made significance advancement in the clinics. However i have few suggestion to improve quality of this articles.

1.Abstract should be formatted objective/method/results/conclusion.

2.Introduction should touch 360 angle of topic.

3.Conclusion need to rewrite for effective delivery of research analysis.

Reviewer #2: Assessing Concentration in the Monoclonal Antibody Innovation Market: A Patent- Based Study

Motta-Santos and colleagues studied the thorough analysis of mAb Market and its implications for healthcare innovation. They highlighted the concentration levels and dissemination patterns of mAb patents

Authors can improve the quality of the manuscript by considering following modifications:

• In the Introduction, authors can highlight the research's novelty and its effectiveness for stakeholders.

• Authors can also discuss the impact of biosimilars on the mAb market.

• Authors can also add some of the case studies or specific examples of successful R&D projects that led to significant changes in mAbs developments or markets.

• Lane 135: Authors can include specific examples of how geographic boundaries influence market dynamics, such as regional variations in drug availability and pricing.

• Authors can discuss about retrieving data for chemical drug patents parallels the process for mAbs with the explanation of major differences.

• Please explain the use of the Herfindahl-Hirschman Index (HHI) and Concentration Ratio (CR). Please add a brief discussion on the significance of these indices in other industries so that readers who are unfamiliar with these concepts understand their relevance.

• The authors provided plenty of data in the table format and text explanation. Authors can consider making bar graphs for the results shown in the table to represent the distribution of patents among holder types for an easy and thorough understanding of the readers. For market concentration, add graphical representations of HHI and CR values.

• Line 358-365: Please explain why is the decrease in participation significant for market dynamics.

• Lines 406–439: Please explain why single-holder contributions dominate and the effect on innovation dynamics. Please explain the barriers to cross-holder partnerships if there are any.

• In Figures 2&3: Please explain why the slope difference matters for market concentration or innovation strategies. What do slopes mean for policy or market behavior?

Reviewer #3: The paper provides a valuable initial analysis of patent concentration in the mAb market. However, it lacks crucial connections between patent data and market realities, a deeper analysis of the implications of concentration, and a broader consideration of key factors shaping the future of the mAb market. Addressing these shortcomings would significantly strengthen the paper's contribution to the field.

As a reviewer, I believe the paper is lacking in several key areas:

1. Lack of correlation between patent concentration and market share:

The paper focuses on patent concentration (HHI based on patents) but fails to connect this to actual market share of the corresponding mAb products.

A crucial aspect is understanding whether the companies with the highest patent concentration also hold the largest market share of approved and marketed mAb drugs.

This would provide a more direct link between patent dominance and potential market power.

2. Limited discussion on the impact of concentration:

The paper mentions "moderate" and "unconcentrated" markets, but lacks a deeper analysis of the potential implications of these concentration levels:

Potential for anti-competitive behavior: Does the level of concentration raise concerns about potential anti-competitive practices like price-fixing or exclusionary behavior?

Impact on innovation: Does the concentration level stifle innovation by hindering entry of new players and limiting access to technology?

Impact on patient access: Does the concentrated market structure potentially lead to higher drug prices and limited patient access to these life-saving therapies?

3. Lack of consideration for therapeutic areas:

The analysis appears to be conducted across the entire mAb market without considering the significant variations within different therapeutic areas (e.g., oncology, autoimmune diseases, infectious diseases).

Concentration levels and their implications can vary significantly across these areas.

Analyzing concentration within specific therapeutic areas would provide a more nuanced understanding of the competitive landscape.

Limited discussion on future trends:

The paper primarily focuses on past trends and current market conditions.

A discussion on potential future trends in the mAb market, such as:

The rise of biosimilars and their impact on market concentration.

The increasing importance of emerging technologies like antibody-drug conjugates (ADCs) and bispecific antibodies.

The potential impact of artificial intelligence (AI) and machine learning on mAb development.

Would enhance the paper's value and provide insights into the evolving competitive landscape.

Reviewer #4: There are no major comments for the article.

1. Authors should also explore factors such as research and development (R&D) investments, collaborations, and licensing agreements that shape the competitive landscape for the mAb market.

2. Add graphical abstract.

6. PLOS authors have the option to publish the peer review history of their article (what does this mean? ). If published, this will include your full peer review and any attached files.

**Do you want your identity to be public for this peer review?** For information about this choice, including consent withdrawal, please see our Privacy Policy .

Reviewer #1: No

Reviewer #2: **Yes: ** PALLAPATI ANUSHA RANI

Reviewer #3: No

Reviewer #4: No

---

## [Author Response · Author response to Decision Letter 1]

31 Jan 2025

Dear reviewers,

We would like to thank you for your careful read of our article and your comments. We truly believe that they help us improve our paper. We addressed all issues appointed by you and provided answers to your questions below.

Best regard,

The authors

REVIEWERS' COMMENTS

Reviewer 1

Comment 1. Abstract should be formatted objective/method/results/conclusion.

A: Thanks for this correction. The abstract was re-structured as requested. Please, find the modified version below.

BACKGROUND: Monoclonal antibodies (mAbs) are revolutionizing healthcare treatments due to their high efficacy and relative safety, despite their cost. Since they first appeared in the late 1980s, a rapidly growing market has developed. OBJECTIVE: This study aims to analyze concentration levels in the market for mAb innovations through a quantitative patent analysis. Data were analyzed using traditional concentration indicators such as the Herfindahl-Hirschman Index and Concentration Ratio, as well as linear regression and kernel density graphs to evaluate innovation and global technology dissemination strategies. The starting point was patents associated with mAbs registered by the FDA and identified in the IQVIA database up until 2019, supplemented by data from The Antibody Society, Purple Book, Orange Book, and FDA. RESULTS: Our findings indicate that the market for mAb innovations is moderately concentrated for general patents and unconcentrated for priority patents. However, it is significantly more concentrated than the market for chemical drug innovations. The mAb patent families tend to generate more progeny patents, although they are deposited in fewer countries. Chemical drug patents spread faster. Some companies seem to be central to the development of mAbs worldwide, including Roche, PDL, City of Hope, and Celltech. Other important players in the mAb innovation market are AbbVie, Amgen, Novartis, GSK, Biogen, BMS, Regeneron, J&J, and AstraZeneca. The most relevant patents in the analysis are associated with methods and procedures to obtain mAbs, not with molecules themselves. CONCLUSION: The concentration in the mAb innovation market is higher than the concentration in the market for chemical drugs innovations. Our findings also indicate that expertise in mAbs development and production is concentrated in a few countries. Additionally, our study identified that a few key players from high-income countries are driving innovation in the mAb market.

Comment 2. Introduction should touch 360 angle of topic.

A: Thank you for this comment. To address it, we included some additional information in the introduction. They include the development of second generation mAb technology and their future potential:

Recently, the development of antibody-drug conjugates (ADCs) and bispecific antibodies has widened even more the usefulness of mAb technology, especially for cancer treatments [17–22]. In the short term, continued investment in R&D for new mAb technologies could provide the world with technologies to treat or cure many rare and orphan diseases [5,6,8,23,24].

A brief history of successful research associated with mAbs and their economic importance for healthcare:

The onset of mAb research happened in 1975, when Milstein and Köhler [25] described a method for obtaining specific antibodies from a continuously growing cell line. In 1986, after some additional advances, the first monoclonal antibody (mAb), Muromonab-CD3, was approved for human use by the Food and Drugs Administration (FDA) [7,14]. However, it was only after the development of the methods for obtaining chimeric, humanized, and human mAbs that the new molecules start getting in the market at an increasing pace [4,26,27]. mAbs are associated with substantial research and development (R&D) costs. According to DiMasi and Grabowski [28], the average capital cost to develop a new biopharmaceutical drug was approximately US$1.2 billion during the 2010s. Despite these significant expenses, mAbs high prices consistently deliver translate into abundant revenues for pharmaceutical companies, making them a lucrative investment [29–31].

An estimates for the concentration on the production market and our focus, that is, the concentration in the innovation market. The last paragraph now reads:

In the pharmaceutical market, three related processes – R&D, production, and commercialization – could be carried out by different companies for the same product, leading to three different estimates of market concentration. Specifically, for the drug production market, there is some evidence on concentration levels [49,50]. They observe that the market as a whole is not concentrated enough to be considered an oligopoly. Craig and Malek [49] found that the five largest pharmaceutical companies retained around 15% of the market in the 1990s [49]. Information on the innovation market, especially mAbs, is much scarcer. This study focuses on R&D by analyzing patent deposit activities, with the specific aim of measuring concentration in the mAbs innovation market using an unprecedent database. Understanding the interactions among players and concentration in the innovation market is essential for developing policies to address the current high prices of mAbs. In addition, the dynamism of the field requires up-to-date analyses. To the best of our knowledge, this study is the first comprehensive analysis of mAbs innovation market.

Some smaller modifications were also introduced to make the introduction clearer and more precise.

Comment 3. Conclusion need to rewrite for effective delivery of research analysis.

A: We modified and removed parts of the conclusion to make it more concise and to the point. Please see the result below:

The main contribution of this paper is the analysis of the concentration within the pharmaceutical innovation market, using a dedicated database to examine the strategies companies employ to maintain market power. Our findings also indicate that expertise in mAbs development and production is concentrated in a few countries, leaving many regions without the necessary capabilities to innovate or manufacture mAbs. Additionally, our study identified key players driving innovation in the mAb market: Roche, PDL, City of Hope, and Celltech. Other significant contributors include AbbVie, Amgen, Novartis, GSK, Biogen, BMS, Regeneron, J&J, and AstraZeneca.

Reviewer 2 - PALLAPATI ANUSHA RANI

Comment 1. In the Introduction, authors can highlight the research's novelty and its effectiveness for stakeholders.

A: Hi. Thank you for this opportunity to include other information to demonstrate the importance of our paper. We included modifications in the last paragraph of the introduction. It now reads:

In the pharmaceutical market, three related processes – R&D, production, and commercialization – could be carried out by different companies for the same product, leading to three different estimates of market concentration. Specifically, for the drug production market, there is some evidence on concentration levels [49,50]. They observe that the market as a whole is not concentrated enough to be considered an oligopoly. Craig and Malek [49] found that the five largest pharmaceutical companies retained around 15% of the market in the 1990s [49]. Information on the innovation market, especially mAbs, is much scarcer. This study focuses on R&D by analyzing patent deposit activities, with the specific aim of measuring concentration in the mAbs innovation market using an unprecedent database. Understanding the interactions among players and concentration in the innovation market is essential for developing policies to address the current high prices of mAbs. In addition, the dynamism of the field requires up-to-date analyses. To the best of our knowledge, this study is the first comprehensive analysis of mAbs innovation market.

Comment 2. Authors can also discuss the impact of biosimilars on the mAb market.

A: Included in the discussion, as requested. Please, see the new paragraph below.

Despite the complications of copying mAbs, the market for biosimilars have been gaining traction over the years. Biological drugs are much more difficult to imitate than chemical drugs. From an economic perspective, the production of biosimilars involves more barriers to entry than the production of small molecules [80]. Specifically, in the case of mAbs, there is an essential differentiation of products. Biosimilars are not generic mAbs. They can be different with respect to the protein sequence and, thus, effects. The expiration of many mAb patents in the next few years should ignite the development of biosimilars. Remicade® (infliximab) sales were slowed by 15% due to biosimilars between 2015 and 2017. A drop that is still modest, considering the prices of reference mAbs. Developing biosimilars that act better than reference products, also known as biobetters or biosuperiors, should accelerate this process [12] and help reduce the concentration in the market for production of mAbs. However, their imitative nature suggests they will not affect innovation market concentration.

Comment 3. Authors can also add some of the case studies or specific examples of successful R&D projects that led to significant changes in mAbs developments or markets.

A: We feel that a good example of a successful R&D project is the Cabilly patents described in the discussion. However, to better comply with this comment, we included a paragraph describing some important research results that helped launched mAbs in the introduction:

The onset of mAb research happened in 1975, when Milstein and Köhler [25] described a method for obtaining specific antibodies from a continuously growing cell line. In 1986, after some additional advances, the first monoclonal antibody (mAb), Muromonab-CD3, was approved for human use by the Food and Drugs Administration (FDA) [7,14]. However, it was only after the development of the methods for obtaining chimeric, humanized, and human mAbs that the new molecules start getting in the market at an increasing pace [4,26,27]. mAbs are associated with substantial research and development (R&D) costs. According to DiMasi and Grabowski [28], the average capital cost to develop a new biopharmaceutical drug was approximately US$1.2 billion during the 2010s. Despite these significant expenses, mAbs high prices consistently deliver translate into abundant revenues for pharmaceutical companies, making them a lucrative investment [29–31].

Comment 4. Lane 135: Authors can include specific examples of how geographic boundaries influence market dynamics, such as regional variations in drug availability and pricing.

A: We appreciate this comment and included a more robust explanation of the geographic boundaries in the pharmaceutical market. Please, find the modification below:

Markets must be defined with respect to their geographic and product dimensions. The geographic dimension comprises the territorial boundaries of the market; i.e. all relevant sources of the product and their spatial disposition. Since the formation of large drug companies in the last century, the borders of pharmaceutical markets have expanded [49]. Despite some trade restrictions between countries imposed by the TRIPS Agreement, the pharmaceutical production market has a global scope in the sense that commercialization of drugs might happen across frontiers [33,52,53]. The same is true for the innovation market. The flow of information occurs across borders through the allocation and enforcement of intellectual property rights (IPRs) in offices in each country. However, patent legislation per se are contained inside the borders of national innovation systems. The dependency of international products subjected to IPR protection presents two challenges for the access to new drugs, especially in LMICs: (i) the monopoly created by the IPR hinders competition and lead to higher prices and (ii) the logistics of distributing a new technology around the world is difficult. Even registering new technologies with local authorities might delay the availability of new drugs.

Comment 5. Authors can discuss about retrieving data for chemical drug patents parallels the process for mAbs with the explanation of major differences.

A: This is a good idea. Please, find the paragraph re-written below:

Figure 1 synthesizes the steps followed to build the database. The first step is the identification of the drugs. The list of mAbs approved by the FDA and their International Non-Proprietary Names (INNs) were collected from The Antibody Society website [57–59]. The Antibody Society is an international non-profit trade association representing individuals and organizations involved in antibody development. The list of INNs of chemical drugs approved by the FDA was collected from the Orange Book [60]. The Orange Book (Approved Drug Products with Therapeutic Equivalence Evaluations) identifies drug products approved by the FDA under the Federal Food, Drug, and Cosmetic Act. The INNs were used to retrieve the US Patent and Trademark Office (USPTO) patent numbers from the IQVIA database, which include data from 130 countries [61].

Comment 6. Please explain the use of the Herfindahl-Hirschman Index (HHI) and Concentration Ratio (CR). Please add a brief discussion on the significance of these indices in other industries so that readers who are unfamiliar with these concepts understand their relevance.

A: To address this issue, two parts of the text were changed. Initially, the significance of the indices were better explained in the methods section:

This paper applies the HHI and the CR (CR4 and CR10) to estimate market concentration. These indices are among the most well-established in the literature. Generally, they evaluate the distribution of market share, measured in our study by patents, among individuals or firms. Therefore, they estimate the competition in an industry [68]. The CR is the cumulative market share of the M largest companies in the market (Equation 1) while the HHI is calculated by summing the squares of the market shares of all holders (Equation 2). Their values range from 0 to 1 (monopoly); more precisely, D_(HHI|CR)=(0,1]. A market can be considered unconcentrated when HHI≤0.10, moderately concentrated between 0.10 and 0.18, and highly concentrated when HHI>0.18 [69–72]. For I companies on a market, the HHI has its lowest value when all players have equal market shares; that is, min(HHI)=1⁄I [68,71]. In the case of the CR, an industry is commonly considered competitive when CR_4<0.4 while CR_4>0.6 is associated with tight oligopolies or a market with a dominant firm with a competitive fringe [68].

A new paragraph was added to the discussion to address this comment as well.

It is paramount to understand how the HHI and CR relate to each other. The relationship between them is not monotonic and depends on the market being evaluated [68]. They do not necessarily indicate the same result, but the analysis of both can lead to some conclusions. One inference that can be drawn from our concentration indices is that the mAb innovation market comprises a core of large companies surrounded by numerous smaller firms; i.e. an oligopoly with a competitive fringe. The high concentration in the mAb innovation market is a significant factor contributing to elevated drug prices. Estimates of HHI and CR4 for different industries are numerous. Each industry has its own specificities and trends. For comparison, some of examples are: the US consumer book industry (HHI=0.118, CR4=0.368) [77], the global personal computers industry (HHI=0.148, CR4=0.707) [78], the Turkish automobile industry (HHI=0.079, CR4=0.462), and the biodiesel industry in Brazil (HHI=0.059, CR4=0.409) [79].

This should be enough for readers unfamiliar with the use of such indices to understand their meaning.

Comment 7. The authors provided plenty of data in the table format and text explanation. Authors can consider making bar graphs for the results shown in the table to represent the distribution of patents among holder types for an easy and thorough understanding of the readers. For market concentration, add graphical representati

---

## [Decision Letter · Decision Letter 1]

26 Feb 2025

Assessing concentration in the monoclonal antibody innovation market: A patent-based study

PONE-D-24-54902R1

Dear Dr. Santos,

We’re pleased to inform you that your manuscript has been judged scientifically suitable for publication and will be formally accepted for publication once it meets all outstanding technical requirements.

Kind regards,

Satish Rojekar, Ph.D.

Academic Editor

PLOS ONE

Reviewers' comments:

Reviewer's Responses to Questions

**Comments to the Author**

1. If the authors have adequately addressed your comments raised in a previous round of review and you feel that this manuscript is now acceptable for publication, you may indicate that here to bypass the “Comments to the Author” section, enter your conflict of interest statement in the “Confidential to Editor” section, and submit your "Accept" recommendation.

Reviewer #2: All comments have been addressed

Reviewer #4: All comments have been addressed

2. Is the manuscript technically sound, and do the data support the conclusions?

Reviewer #2: Yes

Reviewer #4: Yes

3. Has the statistical analysis been performed appropriately and rigorously?

Reviewer #2: Yes

Reviewer #4: Yes

4. Have the authors made all data underlying the findings in their manuscript fully available?

Reviewer #2: Yes

Reviewer #4: Yes

5. Is the manuscript presented in an intelligible fashion and written in standard English?

Reviewer #2: Yes

Reviewer #4: Yes

6. Review Comments to the Author

Reviewer #2: Authors addressed all the comments mentioned in the revision. There are no further comments for this manuscript.

Reviewer #4: Authors have addressed all the comments.

There are no further comments for the revised manuscript.

7. PLOS authors have the option to publish the peer review history of their article (what does this mean? ). If published, this will include your full peer review and any attached files.

**Do you want your identity to be public for this peer review?** For information about this choice, including consent withdrawal, please see our Privacy Policy .

Reviewer #2: **Yes: ** Pallapati Anusha Rani

Reviewer #4: **Yes: ** Kinjal Parikh

---

## [Editor Report · Acceptance letter]

PONE-D-24-54902R1

PLOS ONE

Dear Dr. Motta-Santos,

I'm pleased to inform you that your manuscript has been deemed suitable for publication in PLOS ONE. Congratulations! Your manuscript is now being handed over to our production team.

Kind regards,

on behalf of

Dr. Satish Rojekar

Academic Editor

PLOS ONE